

# Study on The Error Structure of Radar Reflectivity Using The Symmetric Rainrate Predictor

Lidou Huyan[1,2], Yudong Gao[1,2], Zheng Wu[1,2], Bojun Liu[3]

[1]Chongqing Institute of Meteorological Sciences, Chongqing 401147, China.
[2]Chongqing Engineering Research Centre of Agrometeorology and Satellite Remote Sensing, Chongqing 401147, China.
[3]Chongqing Meteorological Observatory, Chongqing 401147, China.

*Correspondence to*: Yudong Gao (stephencool@163.com)

**Abstract.** Given that the Gaussianity of observation error distribution is the fundamental principle of most current modern data assimilation methods, the error structure of radar reflectivity becomes increasingly important with the development of
reflectivity assimilation in convection-allowing numerical weather prediction. This study examines the error distribution of radar reflectivity and discusses what give rise to the non-Gaussian error distribution by using 6 month reflectivity departures between observations and simulations in the Southwest China. By following the symmetric error model in all-sky satellite radiance assimilation, we unveil the error structure of radar reflectivity as a function of symmetric rainrates, which is the average of observed and simulated rainrates. Unlike satellite radiance, the reflectivity error shows a sharper slope in light
precipitations than moderate precipitations. Thus, a three-piecewise fitting function is more suitable for radar reflectivity than a two-piecewise fitting function. The probability distribution functions of reflectivity departures normalized by symmetric rainrates become more Gaussian in comparison with the raw probability distribution function. Moreover, the possibility of using third-party predictor to construct the symmetric error model are also discussed in this study. According to the Jensen-Shannon divergence, a more linear predictor, the logarithmic transformation of rainrate, can provide the most
Gaussian error distribution.

## 1 Introduction

Thanks to the high spatiotemporal resolution of radar data, the assimilation of radar reflectivity using either variational or ensemble-based data assimilation approaches (Sun and Crook, 1997; Snyder and Zhang, 2003; Tong and Xue, 2005; Kong et al., 2018) has shown improvements on the prediction of convective systems (Stensrud et al., 2013; Sun et al., 2014;
Gustafsson et al., 2018). In most current data assimilation methods, only the Gaussian error distribution of observations guarantees statistically optimal estimations. However, few studies have investigated whether the error distribution of radar reflectivity is Gaussian and how to deal with the non-Gaussian error distribution.

The radar echo signal, called reflectivity factor (unit: $mm^6$ $m^{-3}$), is proportional to the sixth power of the hydrometeor diameter according to the Rayleigh scattering. Not only physical aspects of precipitations, but also various characteristics of
the radar itself can complicate the calculation of reflectivity factor. In practice, the equivalent reflectivity (unit: dBZ), which





is a logarithmic transformation of reflectivity factor, is usually assimilated to improve the simulations of convective storms (Jung et al., 2008; Gao and Stensrud, 2012; Wang and Wang, 2017). To attack the convergence problem arisen by the nonlinear observation operator in the minimization process, the logarithmic or power transform on hydrometeor control variables is employed (Liu et al., 2020; Chen et al., 2021). The indirect assimilation of radar reflectivity, retrieving humidity
pseudo-observation from radar reflectivity by a one-dimensional Bayesian inversion (Caumont et al., 2010), is implemented operationally at Météo-France (Martet et al., 2022). Wattrelot et al. (2014) denoted that the error of radar reflectivity resulting from the nonlinear observation operator has significant impacts on the indirect assimilation of reflectivity. Similar to the relative humidity retrieved from radar reflectivity, the assimilation of total column water vapor retrieved from hourly precipitation estimated by radar and rain gauge was about to reduce errors in short-range precipitation forecasts (Lopez and
Bauer, 2007). In 2011, a logarithmic transform of precipitation observations has been directly assimilated in operational four-dimensional variation system at the European Centre for Medium-Range Weather Forecasts (Lopez, 2011).

In earlier studies, the observation error of radar reflectivity was a constant value (Sun and Crook, 1997; Tong and Xue, 2005; Xue et al., 2007). With the popularity of the Desroziers method (Desroziers et al., 2005) in recent years, the spatial error correlations of radar reflectivity were investigated in the Met Office (Waller et al., 2017) and the Deutscher Wetterdienst
(Zeng et al., 2021), but the non-Gaussian error distribution is still a challenge in radar reflectivity assimilation. In this study, we critically examine the non-Gaussian error structure of equivalent reflectivity and attempt to understand what give rise to the non-Gaussian error distribution.

Similar to the satellite radiance in all-sky reported by Geer and Bauer (2011), we can summarize that the radar reflectivity error also exhibits substantial non-Gaussian behaviour because:

1.Boundedness. There are two kinds of boundednesses for radar reflectivity. First, radar reflectivity itself is a bounded variable since the hydrometeors cannot be less than zero. The similar boundedness issue leads to the non-Gaussian error distribution in satellite radiance assimilation. The second boundedness indicates that the radar reflectivity could decrease fast to zero outside the rainy areas, because the distribution of hydrometeors is limited by geophysical boundaries, such as precipitation and non-precipitation areas. Different to satellite radiance assimilation, the discontinuity of hydrometeors in the
background prevents non-precipitation area from assimilating reflectivity. It is called the "zero gradient" effect (Bannister et al., 2020).

2.Heteroscedasticity. The error of equivalent reflectivity can change as a function of precipitation. It is clear in reflectivity assimilation, where errors including representation errors and operator errors increase with the precipitation amount. The representation errors, describing the mismatches between observations and simulations, increase rapidly with model errors
for intense convention, which often exhibit low predictability (Sun and Zhang, 2020). Moreover, the errors of observation operator in reflectivity assimilation also become large when the convective systems intensify. For instance, the simplified reflectivity operator is insufficient to describe the shapes and sizes of ice-phased hydrometeors in strong conventions (Jung et al., 2008).



In an idealized system, Bishop (2019) demonstrated that state-dependent observation error variance should be anticipated
and estimated whenever the observation is of a bounded variable, whose error variance tends to zero as the observation
approaches the bound. Xue et al. (2007) also pointed to the importance of properly modelling reflectivity errors when the
observation operator is nonlinear. The radar reflectivity is distinctly a bounded measurement and has complicated nonlinear
observation operator. As inspired by these previous studies, the error of radar reflectivity should be a state-dependent
function instead of a prescribed constant. In this study, we present the first in-depth study to unveil the error structure of
equivalent reflectivity by following the successful construction of symmetric error model in all-sky satellite radiance
assimilation (Geer and Bauer, 2011; Migliorini and Candy, 2019; Zhu et al., 2019; Shahabadi and Buehner, 2021; Johnson et
al., 2022).

To construct symmetric error model, we need a symmetric predictor, which is the average of simulations and observations.
For radar reflectivity, it should be an estimation of precipitation and can be predicted by numerical weather model. Similar to
the liquid water path derived from satellite radiance observations, the rainrate can be estimated by the radar reflectivity in
terms of the Z-I relationship. Moreover, the radar reflectivity is the main source to produce the stage IV precipitation data at
the National Centres for Environmental Prediction (Nelson et al., 2016), demonstrating that the radar reflectivity is a good
indicator of precipitation. Thus, the rainrate is used as the predictor of the symmetric error model of radar reflectivity to
describe the heteroscedasticity of reflectivity error distribution in this study. It naturally steps forward to examine the effects
of some properties of rainrate on the symmetric error model of radar reflectivity. The accuracy of rainrate observations is the
most uncertain property. It could vary from one data set to another. In this study, we first focus on the effects of observation
accuracy on the symmetric error model. As aforementioned, the logarithmic transform on hydrometeor control variables or
observations can alleviate the nonlinear issue in reflectivity assimilation. Here the linearization is the second property we
attempt to investigate by imposing a logarithmic transform on the rainrate.

The rest of this study is organized as follows. In section 2, observations, model equivalents and their departures are
introduced. Properties of various predictors are discussed in section 3. The error structure of radar reflectivity constructed by
symmetric rainrates is presented in section 4. This section also shows the effects of the accuracy and linearization of
predictor on the symmetric error model of radar reflectivity. Finally, conclusions are given in section 5.

## 2 Observations, model equivalents and their departures

### 2.1 Composite reflectivity observations

To match the three-dimensional reflectivity with the two-dimensional predictor, the composite equivalent reflectivity, which
is the horizontal distribution of vertical maximum radar reflectivity, is used as the observations in this study. Digital mosaics
of composite equivalent reflectivity with 1 km horizontal resolution are operationally provided by the Chongqing
Meteorological Observatory based on the weather radar network in Chongqing Municipality (red circles in Figure 1). This
radar network consists of 5 S band radars and covers the centre and east of the Sichuan Basin. The two black rectangles in



Figure 1 limit the research areas in order to exclude those model results out of the radar coverage. The 6 month composite equivalent reflectivities (hereafter shorted by reflectivities, unit: dBZ), from April to September in 2021, are used to investigate the symmetric error model of radar reflectivity.

Figure 2a gives reflectivity distribution of a rainstorm captured by the radar network at 0200 UTC on May 3th 2021. The reflectivity indicates clear geophysical boundaries of the rainy areas and storms. Although reflectivity is not exactly identical to precipitation, we attempt to choose rainrate as the predictor of symmetric error model because both reflectivity and rainrate are good indicators of convective storms. In this study, reflectivities less than 5 dBZ in either the observations or the model equivalents, representing 'non-rainy' areas, were excluded in order to match the rainrate predictor.

## 2.2 Model equivalents

The 6 month model equivalents of reflectivities are simulated by the Weather Research and Forecasting (WRF; Skamarock et al., 2019) model Version 4.1. The Lambert projection, whose standard latitudes are 20° N and 30° N and standard longitude is 106.5° E, is used. Same physics packages, including the new Kain-Fritsch scheme (Kain, 2004), the Yonsei University planetary scheme (YSU) (Hong et al., 2006), the Thompson scheme (Thompson et al., 2008) and Unified Noah Land Surface Model (Ek et al., 2003), are employed in the 6 month simulations. The WRF model has been nested in one-way with a coarse resolution of 9 km and a fine resolution of 3 km. Figure 1 gives the topography in the inner domain of WRF model, whose central location is at (29.8° N, 106.58° E) and horizontal grids are 480×360. In the outmost domain, the central location is at (30° N, 104.5° E) and the horizontal grids are 600×480. The two domains have 51 vertical layers.

The initial and lateral boundary conditions of the WRF model are 0.5°×0.5° Global Forecast system (GFS) data sets produced by the National Centres for Environmental Prediction (NCEP). More information about GFS data sets is available at https://www.ncdc.noaa.gov/data-access/model-data/model-datasets/global-forecast-system-gfs. The GFS analyses at 0000 UTC and 1200 UTC in the 6 months are used to drive the WRF model. The model equivalents are computed using 6 hour simulations, because a shorter simulation time causes spin-up issues and a longer simulation time brings large model errors. In this study, the model equivalents have 12 hour time interval (i.e., 0600 UTC and 1800 UTC) from April to September in 2021.

The diagnostic algorithm of model equivalents is from the Unified Post Processor (UPP) package (https://dtcenter.org/community-code/unified-post-processor-upp), which is recommended to compute the simulated reflectivity from WRF model (Min et al., 2015). The three-dimensional diagnostic reflectivity on model levels, consisting of rain drops, snow particles and graupel particles, can be briefly described as:

$$Z = 10 \log_{10}(Z_{er} + Z_{es} + Z_{eg})$$

where $Z_{er}$, $Z_{es}$ and $Z_{eg}$ are reflectivity factor for rain, snow and graupel, respectively. More details of this diagnostic algorithm, including the densities and intercept parameters, can be found in Stoelinga (2005). The UPP interpolates model equivalents from the coordinates of WRF model to standard pressure levels and then generates the model equivalents of composite equivalent reflectivities.



The excessive model equivalents outside the radar network coverage could increase mismatches between observations and
simulations because the truth outside the radar network is unknown (shaded by crosses in Figure 1). In this study,
constructing the symmetric error model of reflectivity is restricted to local areas (two black rectangles in Figure 1), which is
approximately identical to the radar network coverage. To match with the rainrate resolution, we interpolated linearly
reflectivities and model equivalents to 5 km resolution.

In Figure 2b, the model equivalents are incorrectly located, poorly shaped, or have inaccurate intensity in comparison with
observations. Followed by Geer and Bauer (2011), we also refer all these errors to 'mislocation' error. The mislocation error
of reflectivities can vary widely from place to place, implying that a constant standard deviation may be insufficient to
describe the error structure of reflectivity in data assimilation. Because the mislocation error often appears to be much worse
at convective scale than synoptic scale according to the comparison of probability distribution functions (PDFs) in the
following subsection, reflectivity assimilation may need a more elaborate error structure in convective-allowing resolution.
Similar to radar observations, the model equivalents also have geophysical boundaries in the background. The zero gradient
of hydrometeors resulted from geophysical boundaries impedes assimilating reflectivity.

The mislocation error can result in the non-Gaussian error distribution that violates the fundamental principle underlying
most current assimilation methods. The aim of the symmetric error model is to describe the standard deviation of reflectivity
as the function of rainrate and construct a near Gaussian error distribution assigning appropriate weight on reflectivity
assimilation.

## 2.3 Departures between observations and model equivalents

Figure 3a shows a histogram of the reflectivities against the model equivalents based on a sample of 1787233 observations,
using all reflectivities larger than 5 dBZ (i.e., including the misses and false simulations). Similar to the behaviours of
satellite radiances in cloudy sky (Geer and Bauer, 2011; Migliorini and Candy, 2019), the wide spread of reflectivities
confirms that the model equivalents are often very different from observations in 6 month simulations. However, the high
numbers along the abscissa and ordinate imply the large mislocation errors of reflectivities resulting from considerable
misses and false simulations. By comparing with the satellite radiance departures (Figure 5 in Migliorini and Candy, 2019),
these considerable mismatches are associated with worse spatial discontinuities in reflectivity departures between
observations and simulations. Thus, the non-Gaussian error distribution in radar reflectivity assimilation is likely to be
stronger than that in satellite radiance assimilation. For convenience we refer to the discontinuous scenario as 'any-
reflectivity'.

To examine effects of the large mislocation errors on the reflectivity error structure, we removed all misses and false
simulations in observations and model equivalents and obtained 540529 observation samples (Figure 3b). We refer to this
scenario as 'both-reflectivity', whose histogram is similar to the nonprecipitating cloud affected satellite radiance observed
by the AMSR-E channel 37v (Geer and Bauer, 2011). It could be interpreted as Figure 3 showing that the reflectivity in
'any-reflectivity' has a more complicated error structure than 'both-reflectivity'.



The departures between observations and model equivalents are one of the main sources of information on the observation error structure. It is important to examine the raw PDFs of 'any-reflectivity' and 'both-reflectivity' departures before constructing the symmetric error model. In Figure 4, the PDF of 'any-reflectivity' shows that a bimodal distribution denotes

the numerous misses (the right peak of red line) and false simulations (the left peak of red line), which gives an undesirable effect on reflectivity assimilation. Instead, the PDF of 'both-reflectivity' departures displays a unimodal distribution. Although the unimodal distribution is not a Gaussian distribution, it is closer to the Gaussian distribution than that of 'any-reflectivity' departures. It also illustrates that the error structure is more complicated in 'any-reflectivity', where the misses and false simulations are included, than that in 'both-reflectivity'.

## 3 Predictors of symmetric error model


### 3.1 Predictor derived from reflectivity

The predictors of previous symmetric error models in satellite radiance assimilation were derived from the satellite radiance observations. Similarly, the rainrate can be derived from the echo signal in terms of the Z-I relationship, which is an empirical formula that is used to estimate rainrate I (unit: mm h$^{-1}$) from reflectivity factor $Z_e$ (unit: mm$^6$ m$^{-3}$):

$$Z_e = aI^b \tag{1}$$

Here, the reflectivity factor at 3 km altitude and typical coefficients a=300 and b=1.4 are employed. Therefore, the 'symmetric' rainrate, $rr_{sym}$, which is used as the symmetric predictor in this study, is the average of derived rainrate, $rr_{obs}$, and simulated rainrate, $rr_{model}$:

$$rr_{sym} = 0.5 \times (rr_{obs} + rr_{model}) \tag{2}$$

In this study, the $rr_{model}$ is the rainrate simulated by WRF, not derived by the reflectivity simulation.

Figure 5 shows the distributions of various rainrates data sets at the same time in Figure 2. In Figure 5a, some small heavy rainfall centres disagree with weak reflectivity observations, but the largest rainy area derived from reflectivity factor well corresponds to the strong radar observations in Figure 2a. Thus, the derived rainrate is the comparable measurement of radar reflectivity, despite some overestimated rainrates coexist with moderate reflectivities. Although some non-precipitation areas

exist in reflectivity areas, the rainrates derived from reflectivity factors present similar geophysical boundaries of non-precipitation and precipitation, chosen to be 0.1 mm h$^{-1}$ in this study. In the comparison of derived rainrates and simulated rainrates (Figure 5d), the mislocation errors of the rainrates are changed widely from place to place, similar to those of reflectivities, illustrating that the error structure of reflectivities associated with the characteristics of convective systems can be described by the rainrate predictor.





## 3.2 Predictors from third-party observations


Derivation from reflectivity factor is not the only way to obtain the rainrate data sets. Other hourly precipitation observations can be used to compute rainrate measurements. Thus, it is of interest to discuss how the accuracy of predictor affects the symmetric error model by comparing the differences between derived predictor and third-party predictors.

In this study, the derived rainrates are replaced by another two data sets, including the CMA Multisource Precipitation

Analysis System (CMPAS) data sets provided by National Meteorological Information Center of the China Meteorological Administration (NMIC/CMA) and the FY-4A Quantitative Precipitation Estimation (QPE) products provided by the National Satellite Meteorological Center of the China Meteorological Administration (NSMC/CMA; http://satellite.nsmc.org.cn/). The hourly CMPAS data sets with 0.05º resolution, merging precipitation observations from rain gauge, radar QPE and satellite QPE, capture a number of details of hourly precipitations and are more accurate than

other single source precipitation observations (Pan et al., 2018; Li et al., 2022). The hourly FY-4A QPE products with 4 km resolution offer critical information for monitoring and forecasting severe storms (Yang et al., 2017). The quality control of QPE products has been performed by NSMC/CMA. We used the QPE products labelled 'perfect' and 'good' in this study. To match with the radar observations and model equivalents, the CMPAS data sets and FY-4A QPE products are interpolated linearly to 5 km resolution in the same local areas (the black rectangles in Figure 1). The CMPAS and FY-4A

rainrates, the average of hourly precipitation amounts at two consecutive hours, are used to examine the effects of observation accuracy on the symmetric error model.

As shown in Figure 5b, the rainy areas indicated by CMPAS rainrates are similar to the rainrates derived from reflectivity factors because the radar QPEs have been used to generate the CMPAS data sets. For instants, the largest rain belt locates from 28º N to 30º N in both rainrate data sets. Therefore, the CMPAS rainrates are also comparable to reflectivity

observations especially for heavy precipitations. Although there are perceptible differences between CMPAS rainrates and derived rainrates, the differences between CMPAS rainrates and simulated rainrates are similar to those between derived rainrates and simulated rainrates. It illustrates that the error distributions of derived rainrates and CMPAS rainrates could have same substantial features. However, the FY-4A rainrates lose a number of detailed structures in the rainiest area instead of a homogeneous and outspread heavy rainy area (Figure 5c). Owing to additional errors, building the symmetric error

model with a less accurate predictor could provide a worse description of the observation errors.

In order to examine the relationship between reflectivity departures and symmetric rainrates, it is advisable to count the numbers of reflectivity departures over the discrete intervals of symmetric rainrates, chosen here to be 0.5 mm h$^{-1}$. Figure 6a shows the major 'any-reflectivity' departures, chosen to be larger than 100 samples, become negative as symmetric rainrate increases roughly from 0.1 to 10 mm h$^{-1}$, suggesting that the WRF model produced excessive reflectivities in the 6 month

simulations. The values in major departures are roughly from -50 to 30 dBZ. For 'both-reflectivity' departures (Figure 6b), the major departures span from -40 to 25 dBZ because misses and false simulations are removed. The negative bias of major 'both-reflectivity' departures declines as well. In the comparison of Figure 6a and Figure 6b, the distribution of 'both-





reflectivity' departures is more similar to the relationship between the satellite radiance departures (Migliorini and Candy, 2019; Zhu et al., 2019; Shahabadi and Buehner, 2021). It could be argued that the larger mislocation error at convective scale

is a challenge for the direct assimilation of reflectivity owning to the stronger non-Gaussianity.

Figure 7 shows the numbers of 'any-reflectivity' departures over various predictors. For the CMPAS data sets (Figure 7a), the major departures show similar values, roughly from -50 to 30 dBZ, to the 'any-reflectivity' departures (Figure 6a). However, the negative bias is similar to the 'both-reflectivity' departures (Figure 6b). We argue that the accurate rainrate could enhance the representation of symmetric predictor in the construction of symmetric error model.

For the FY-4A data sets (Figure 7b), the 'any-reflectivity' departures spread more widely and homogeneously and lose the rapid variation roughly over 0.1 to 3 mm h$^{-1}$ by comparing with the derived rainrates (Figure 6a). It may be consistent with the homogeneous distribution of FY-4A rainrates (Figure 5c). Moreover, the 'any-reflectivity' departures in Figure 7b exhibit a relatively large number at around 10 mm h$^{-1}$, which may be the sampling error coming from the quality of FY-4A data sets.

### 3.3 The linearization of predictor

The Z-I relationship exists between rainrate and reflectivity factor $Z_e$ (unit: mm$^6$ m$^{-3}$), not equivalent reflectivity Z (unit: dBZ). A natural step forward is imposing a logarithmic transformation on Eq. 1 in order to obtain a more linear relationship between equivalent reflectivity and symmetric predictor:

$$Z = 10 \log_{10} Z_e = 10 \log_{10} a + 10 b \log_{10} I \qquad (3)$$

where a and b are the coefficients of Z-I relationship. In this study, the Eq. 3 is not a formula to obtain the quantitative equivalent reflectivity accurately. It merely transforms the observation and predictor to a more linear space, which allows us to discuss the effects of the linearization of predictor on the symmetric error model. The histogram of 'any-reflectivity' departures against symmetric rainrates computed by the 10 times logarithmic rainrates (i.e., $10 \log_{10} I$ in Eq. 3) is showed in Figure 7c. Because the logarithmic transformation requires a value larger than zero, some zero rainrate samples in either the

derived observations or the model simulations are removed. Although the total sample number decreases to 178012 after using the logarithmic transformation, the Figure 7c is much more symmetric about the ordinate than Figure 6a owing to the better representation after removing the zero rainrates in either derived observations or the model simulations. Meanwhile, the Figure 7c shows that the major departures spread widely along the abscissa in comparison with the non-linear symmetric predictor. The distribution of 'any-reflectivity' departures varying with the symmetric logarithmic rainrates becomes very

gentle owing to the logarithmic transformation.



## 4 Reflectivity errors as a function of symmetric rainrates

### 4.1 The symmetric error model of reflectivity

Similar to the satellite radiances, it is possible to investigate the error structure of radar reflectivity over the discrete rainrate bins, chosen to be 0.5 mm h$^{-1}$ in this study. Figure 8a shows that the standard deviations of 'any-reflectivity' departures

increase with symmetric rainrates before peaking at 15 mm h$^{-1}$. The standard deviations grow faster from 0.5 to 1.5 mm h$^{-1}$ than that does from 1.5 to 9.0 mm h$^{-1}$. To illustrate this, we may argue that the light precipitation is closer to the geophysical boundary than the moderate precipitation, resulting in a sharper slope over the first three bins. In contrast to the symmetric error models of satellite radiance, the standard deviations still increase after 9.0 mm h$^{-1}$, illustrating that the WRF model produces poor forecasts of intense rainstorms at convective scale owing to errors of initial conditions and numerical weather

model.

The variation of standard deviations indicates that the error structure of 'any-reflectivity' can be a function of symmetric rainrates. However, the reflectivity error is often a prescribed constant, such as 3 or 5 dBZ, in most researches and operational assimilation systems. This may overestimate the weight of radar reflectivity in data assimilation, especially for intense convective systems. As shown in Figure 8a, the standard deviations of 'any-reflectivity' could vary from about 12 to

35 dBZ. A constant value is insufficient to describe the error structure of radar reflectivity.

To simplify the complex error structure of 'any-reflectivity', we fitted three-piecewise (blue dash line) and two-piecewise (red dash line) functions of symmetric rainrates by using linear regression. A straight line rather than the linear regression is used to describe the reflectivity error for large symmetric rainrates. It is a cautious approach to avoid fitting a false linear regression based on a small sample size, chosen to be smaller than about $10^3$ samples. The first rows in Table 1 and Table 2

list parameters of piecewise functions of 'any-reflectivity'. By comparing with the two-piecewise function, the three-piecewise function has larger slope and smaller RMSE from 0.5 to 9.0 mm h$^{-1}$. It illustrates that the three-piecewise function carry more detailed information about the reflectivity error structure than the two-piecewise function.

Figure 8b shows that the standard deviations of 'both-reflectivity' departures. The standard deviations of 'both-reflectivity' are smaller than those of 'any-reflectivity', especially before 9.0 mm h$^{-1}$. Because a number of misses and false simulations

are removed. Compared with 'any-reflectivity', the RMSEs of linear regressions for 'both-reflectivity' are improved regardless of whether three-piecewise or two-piecewise functions are fitted (Table 1 and Table 2). The differences of linear regression functions between 'any-reflectivity' and 'both-reflectivity' are small, illustrating that the symmetric error models of 'any-reflectivity' and 'both-reflectivity' are resemble each other closely. It confirms that the PDFs of 'any-reflectivity' and 'both-reflectivity' departures have similar distributions normalized by the symmetric rainrates despite their raw PDFs

are very different.

The standard deviations of 'any-reflectivity' departures constructed by the CMPAS predictor captures similar essential characteristics to that constructed by predictor derived from reflectivity factors (Figure 9a). However, the standard deviations normalized by CMPAS predictor decrease when the symmetric rainrates are larger than 10 mm h$^{-1}$. In contrast to the





predictor derived from reflectivity factors (Figure 8a), the CMPAS predictor shows better agreements between simulations
and observations for large symmetric rainrates. The more accurate predictor only effects on the tail of the symmetric error
model where the variation of standard deviations is omitted, but the RMSEs and correlations of linear regressions are
improved in comparison with the predictor derived from reflectivity factors (the third rows in Table 1 and Table 2).

For FY-4A predictor (Figure 9b), the differences of slopes between three-piecewise and two-piecewise functions are very
small, resulting from the homogeneous distribution of 'any-reflectivity' departures as Figure 7b shows. Although the RMSEs
of linear regressions by using FY-4A predictor are smaller than those by using the derived predictor, the standard deviations
of 'any-reflectivity' departures lose the rapid growth from 0.5 to 1.5 mm h$^{-1}$ rainrates, which is a nontrivial feature of
symmetric error model of reflectivity at convective scale. It is obvious that considerable additional errors can deform the
symmetric error. However, how to quantify the effect of independent errors on the symmetric error model is in need of future
studies.

Figure 9c shows that the standard deviations of 'any-reflectivity' departures increase gently from 9 to 17.5 dBZ when the
symmetric logarithmic rainrates increase from -23 to 6. Only two-piecewise function can be fitted after using the logarithmic
transformation. The linear regression constructed by the logarithmic predictor has the smallest slope among symmetric error
models in Table 2, conforming that the logarithmic transformation removes the large gradient of 'any-reflectivity' departures
over low symmetric logarithmic rainrates as shown in Figure 7c. Meanwhile, the linear regression of the logarithmic
predictor exhibits the highest correlation and the smallest RMSE in Table 2.

**4.2 Improvements on Gaussianity**

To illustrate the potential benefits of symmetric error models to reflectivity assimilation, we examined the Gaussianity of
PDFs in this subsection. Using the binned standard deviations and piecewise functions from Figure 8 and Figure 9, we can
normalize reflectivity departures by using the functions of symmetric predictors. Figure 10 shows the PDFs of 'any-
reflectivity' and 'both-reflectivity' departures normalized by symmetric rainrates, with the raw and normal Gaussian PDFs
for comparison.

Both two-piecewise and three-piecewise functions change the raw PDF of 'any-reflectivity' departures more Gaussian
(Figure 10a). However, the differences between two-piecewise and three-piecewise functions are distinct, illustrating that a
more complicated error model is necessary to correct the non-Gaussian error distribution of radar reflectivity at convective
scale. We argue that the three-piecewise function is sufficient in this study because it shows an identical PDF to binned
standard deviations.

To quantify the similarity between the PDFs normalized by the symmetric rainrates and normal Gaussian PDF, Table 3 lists
the Jensen-Shannon divergence (JSD):

$$\mathrm{JSD}(P \parallel Q) = \frac{1}{2}\sum P(x) \log(\frac{2P(x)}{P(x)+Q(x)}) + \frac{1}{2}\sum Q(x) \log(\frac{2Q(x)}{P(x)+Q(x)}) \qquad (4)$$



where P is the PDFs normalized by symmetric rainrates or raw standard deviations and Q represents the normal Gaussian
        PDF. The JSD is zero means distributions P and Q are the same. For 'any-reflectivity', the JSDs of PDFs normalized by two-
        piecewise and three-piecewise functions decrease to 1.024 and 0.567 respectively from 2.482 normalized by the standard
        deviation of the whole sample.

        The Gaussianity of 'both-reflectivity' PDF normalized by the three-piecewise function is improved (Figure 10b). However,
the differences between JSDs are very small even the symmetric error model are used (Table 3), declined from 0.263 to
        0.137. It illustrates that the symmetric error model only has a little effect on reflectivity departures if the misses and false
        simulations are excluded. In practice, the misses and false simulations should be included in reflectivity assimilation. The
        non-Gaussian distribution of reflectivity error results from the large mislocation errors in the main. Thus, it could be argued
        that constructing an approximate Gaussian error model by symmetric rainrates is helpful in reflectivity assimilation. Instead,
the symmetric error model may be unnecessary for observations whose error distribution is almost Gaussian.

        In comparison with the derived predictor, the PDFs of CMPAS predictor draw similar results (Figure 11a), even though the
        CMPAS predictor is more accurate. Because the derived predictor and CMPAS predictor have similar fitting functions as
        shown in Table 1 and 2. For FY-4A predictor, the differences of PDFs between two-piecewise and three-piecewise functions
        are negligible (Figure 11b). Figure 11c shows the logarithmic transformation changes the raw PDF from bimodality to
unimodality with a little skewness. In Table 3, The JSD of symmetric logarithmic predictor is the smallest among PDFs for
        'any-reflectivity' departures, suggesting that the symmetric logarithmic rainrates achieve the most Gaussian PDF. By
        comparing with the nonlinear predictor derived from reflectivity, the symmetric error model of 'any-reflectivity' is improved
        by the more linear predictor.

**5 Conclusions**

In this study, we used 6 month composite equivalent reflectivities covering local areas in the Southwest China to examine
        the non-Gaussian error structure of radar reflectivity. The non-Gaussianity of reflectivity error is arisen by mislocation errors
        for the most part. The mislocation errors include incorrect location, poor shape and inaccurate intensity between simulations
        and observations and can vary widely from place to place at convective scale, confirming that a constant value for
        observation error is insufficient to describe the error structure of reflectivity in data assimilation, especially in the
convective-allowing resolution. Removing misses and false simulations changes the PDF of raw reflectivity departures more
        Gaussian, which is similar to the nonprecipitating cloud affected satellite radiance.

        We attempted to construct a symmetric error model, which has been broadly used in satellite radiance assimilation
        (Migliorini and Candy, 2019; Zhu et al., 2019; Shahabadi and Buehner, 2021), to describe reflectivity errors as the function
        of symmetric rainrates. In this study, the standard deviations of reflectivity could vary from about 12 to 35 dBZ according to
the symmetric error model constructed by the rainrate predictor. Yet the instrument noise of radar is of order 1 dBZ. Similar
        to satellite radiance, the departure of radar reflectivity between observation and background increases with the symmetric



rainrate amount, illustrating that the largest component of the reflectivity error comes from poor predictions of clouds and rains and inaccurate observation operator in reflectivity assimilation. As the discussion in Geer and Bauer (2011), the inflated reflectivity error may also compensate for an inadequate specification of hydrometeors in background error, which

will be investigated by data assimilation experiments in our ongoing study. In contrast to satellite radiance error, the reflectivity error shows a sharper slope in light precipitations than moderate precipitations, which corresponds to a more complicated structure of reflectivity error at convective scale. By comparing with the PDF of reflectivity departures normalized by the raw standard deviation, the PDFs normalized by symmetric rainrates become more Gaussian, changing a bimodal distribution to a unimodal distribution. Thus, the rainrate-dependent reflectivity error corresponds well to the

principle of most current data assimilation methods.

Effects of the third-party symmetric predictors on the error structure of radar reflectivity are also examined in this study. The CMPAS and FY-4A data sets are used to obtain a more and less accurate predictor respectively. Although the CMPAS predictor is a more accurate predictor, it exhibits similar essential characteristics to the derived predictor in symmetric error models. However, the symmetric error model constructed by the FY-4A predictor loses the rapid growth over small

symmetric rainrates because of the poor representation of FY-4A rainrates, especially at convective scale. Owing to considerable additional errors of FY-4A rainrates, building the symmetric error model with a less accurate predictor provides a worse description of the error structure. Although we cannot quantify a threshold for additional errors to decide whether use the third-party predictor or not, it may broad the usage of the symmetric error model in data assimilation.

The linearization of predictor gives profound effects on the symmetric error model of radar reflectivity. The logarithmic

transformation on rainrates is used to improve linearity of the predictor. Not only the gradients of reflectivity departures as a function of rainrate predictor become gentle, but the PDFs normalized by logarithmic predictor also obtain the smallest JSDs among the PDFs for 'any-reflectivity' departures. It is convenient to create configuration files for the logarithmic predictor in the operational system. Thus, using a more linear predictor is recommended to construct the symmetric error model of radar reflectivity.

In theory, the error structure of radar reflectivity constructed by symmetric rainrates could improve the analysis since the Gaussian error distribution is the fundamental principle in the most current data assimilation systems. Performing a number of experiments to discuss the effects of symmetric error model on reflectivity assimilation is encouraged in future studies. It is worth noting that this study builds a one-dimensional symmetric error model as shown in Figure 8a by using two-dimensional composite reflectivity and rainrate data sets. In future, how to choose a three-dimensional predictor to describe

the variation of reflectivity errors in three-dimensional space is still a challenge. Probably the polarized measurements and their combinations that provide additional information about hydrometeors could be the candidates. It is also of interest to address the insensitivity of symmetric error model to additional errors associated with numerical weather model and observational operator to some extent.



**Code and data availability**

The Observation minus Background data and rainrate data are available at https://doi.org/10.5281/zenodo.7332495. The graphics were generated using NCAR Commend Language (https://www.ncl.ucar.edu/Download/). The Weather Research and Forecasting (WRF) Model (V4.1) used in this study is available from the public WRF-Model Release page on GitHub (https://github.com/wrf-model). The Unified Post Processing System (UPP) for WRF is also available at GitHub (https://github.com/NOAA-EMC/UPP).

**Author contribution**

YG conceptualized this study. LH and YG computed the Observation minus Background data sets and built the error model of radar reflectivity. WZ performed the WRF simulations and BL implemented quality control for radar reflectivity. LH plotted all figures. YG prepared the paper and its revised versions with contributions from all authors.

**Competing interests**

The authors declare that they have no conflict of interest.

**Acknowledgements**

This study was jointly funded by National Key Research and Development Program of China (2017YFC1502001), Natural Science Foundation of Chongqing (cstc2021jcyj-msxmX0698), Innovation and Development Project of the China Meteorological Administration (CXFZ2022P017) and Technical Project of Chongqing Meteorological Bureau (YWJSGG-

395    202212).

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





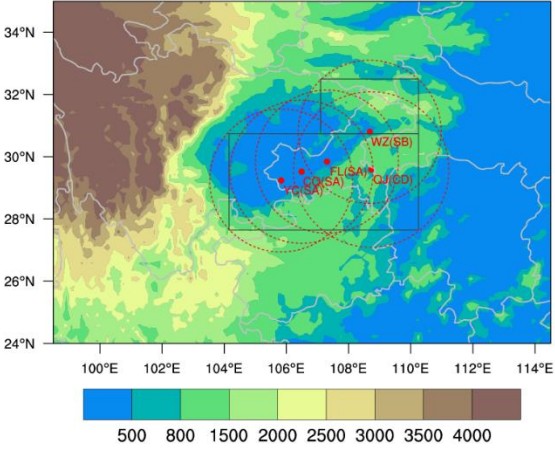

**Figure 1: The inner domain and its topography (shaded; units: m) of WRF model. The red dots and red dash circles denote radar stations and the coverage of radar network respectively. The research areas are limited by the black rectangles to exclude areas that are not covered by radar network.**

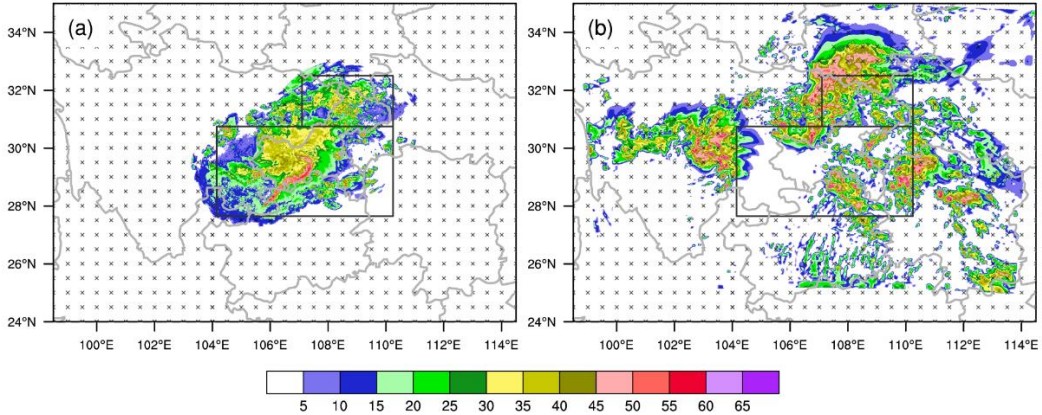

**Figure 2: Distributions of composite equivalent reflectivity (unit: dBZ) (a) observed by radars and (b) their model equivalents at 0200 UTC on May 3th, 2021. The black rectangles indicate the research areas as Figure 1. The crosses show that areas are excluded in computation of reflectivity departures.**



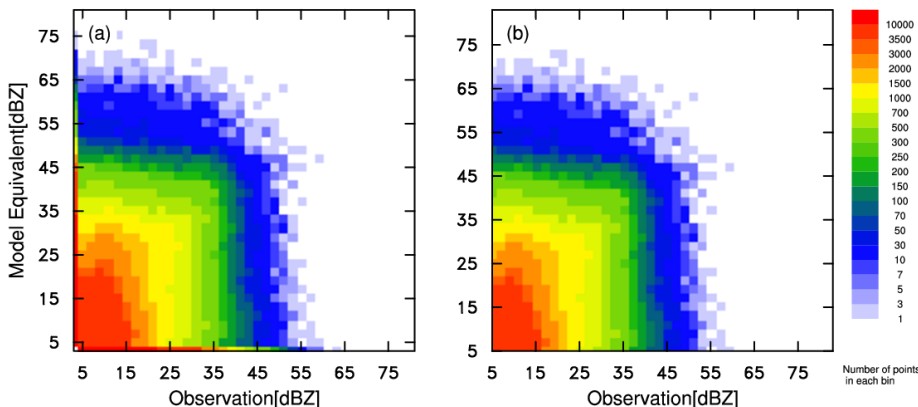

**Figure 3: Histograms for the comparison of (a) 'any-reflectivity' and (b) 'both-reflectivity' observations (abscissa, unit: dBZ) against model equivalents (ordinate, unit: dBZ). The 'any-reflectivity' and 'both-reflectivity' represent that samples include and exclude misses and false simulations, respectively.**

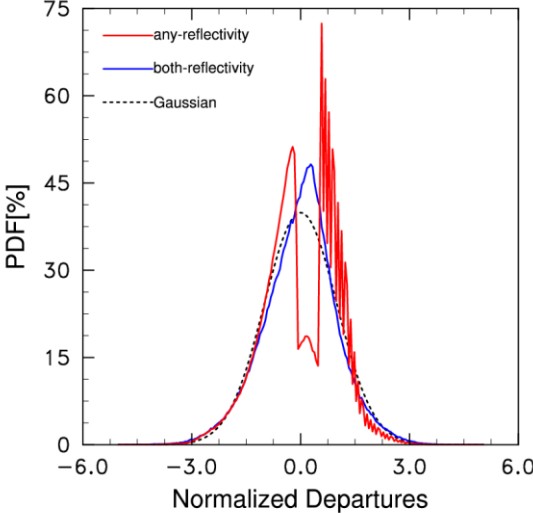

**Figure 4: Probability distribution functions (PDFs) of 'any-reflectivity' (red solid line) and 'both-reflectivity' (blue solid line) departures, normalized by the standard deviation of the whole sample. The black dot line represents the normal Gaussian distribution.**

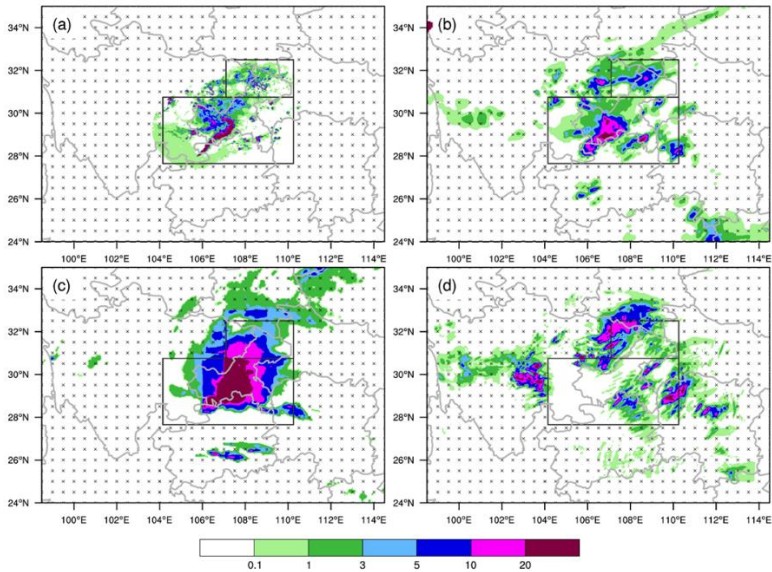

**Figure 5: Distributions of rainrates (unit: mm h⁻¹) (a) derived from reflectivity factors (unit: mm⁶ m⁻³), (b) computed by CMPAS**
**precipitation (unit: mm), (c) computed by FY-4A QPE (unit: mm) and (d) simulated by WRF model at the same time in Figure 2.**
**The black rectangles indicate the research areas as Figure 1. The crosses show the areas are excluded in computation of reflectivity**
**departures.**

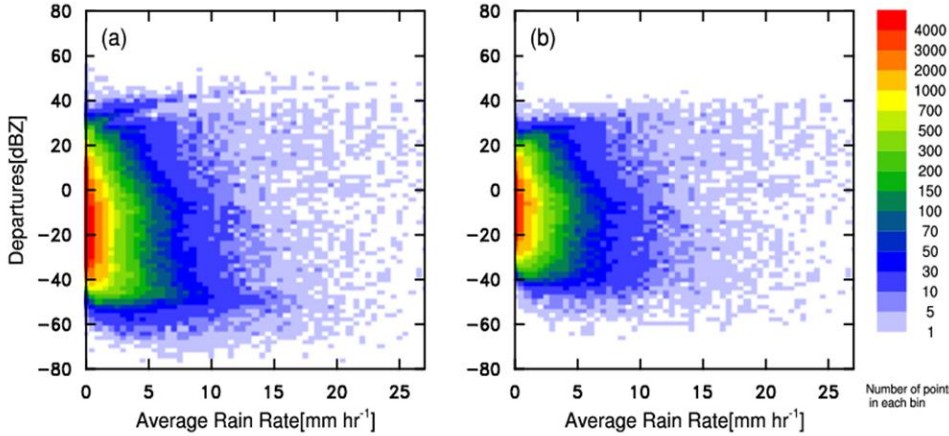

**Figure 6: Histograms of (a) 'any-reflectivity' and (b) 'both-reflectivity' departures (ordinate, unit: dBZ) against symmetric**
**rainrates (abscissa, unit: mm h⁻¹) derived from reflectivity factors.**





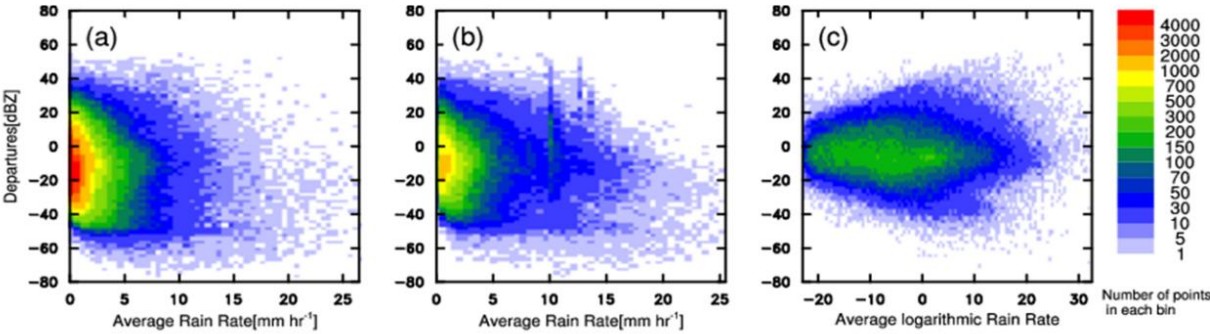

**Figure 7: Histograms of 'any-reflectivity' departures (ordinate, unit: dBZ) against different symmetric rainrates (abscissa, unit: mm h⁻¹), which are computed by (a) the CMPAS, (b) the FY-4A data sets and (c) the 10 times logarithmic rainrate derived from reflectivity factors, respectively.**

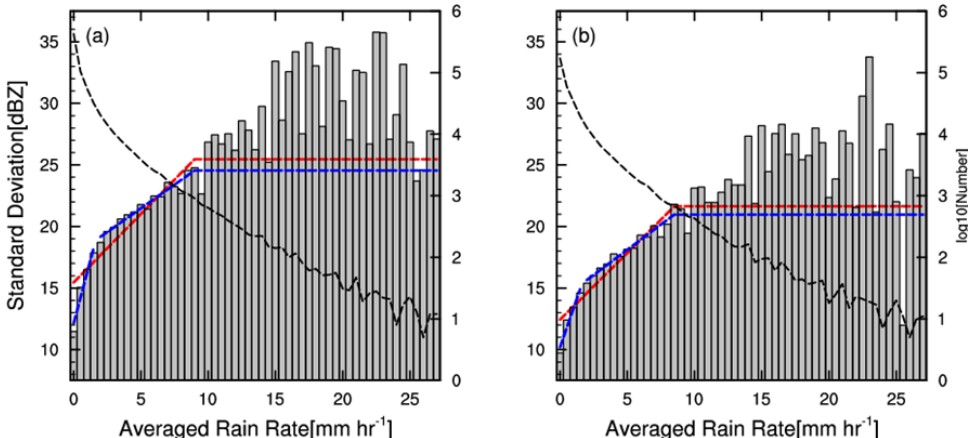

**Figure 8: Standard deviations for (a) 'any-reflectivity' and (b) 'both-reflectivity' departures (unit: dBZ) over symmetric rainrate (unit: mm h⁻¹) bins of size 0.5 mm h⁻¹, derived from reflectivity. The blue and red dash lines show the three-piecewise and two-piecewise linear regressions of symmetric rainrates, repressively. The black dash line shows the logarithm of sample numbers over symmetric rainrate bins.**





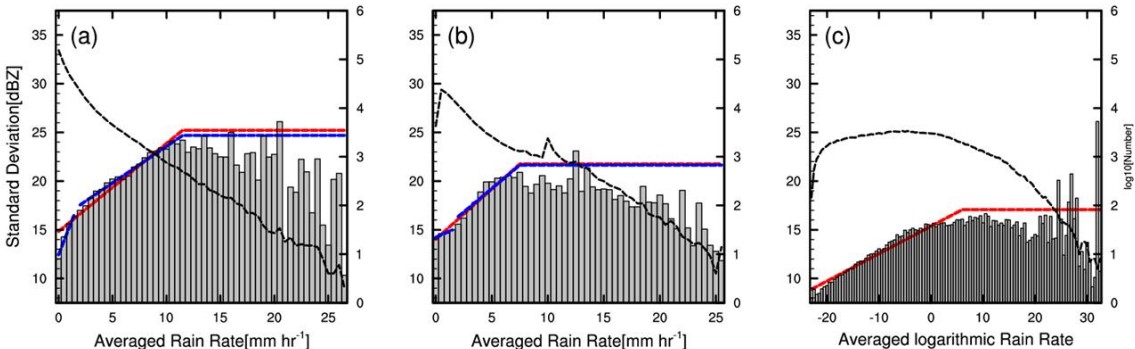

**Figure 9: Standard deviations of 'any-reflectivity' departures (unit: dBZ) over symmetric rainrates (unit: mm h⁻¹) computed by (a) the CMPAS, (b) the FY-4A data sets and (c) the 10 times logarithmic rainrate derived from reflectivity factors, respectively. The black, blue and red dash lines are the same as Figure 8.**

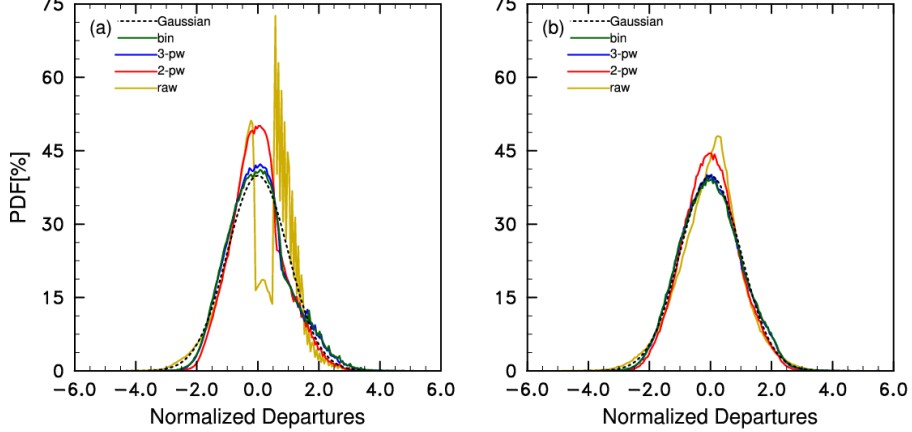

**Figure 10: Probability distribution functions (PDFs) of (a) 'any-reflectivity' and (b) 'both-reflectivity' departures normalized by symmetric rainrates which are computed by rainrates derived from reflectivity factors. The yellow, red, blue, green and black lines represent the raw, two-piecewise, three-piecewise, binned and normal Gaussian PDFs, respectively.**





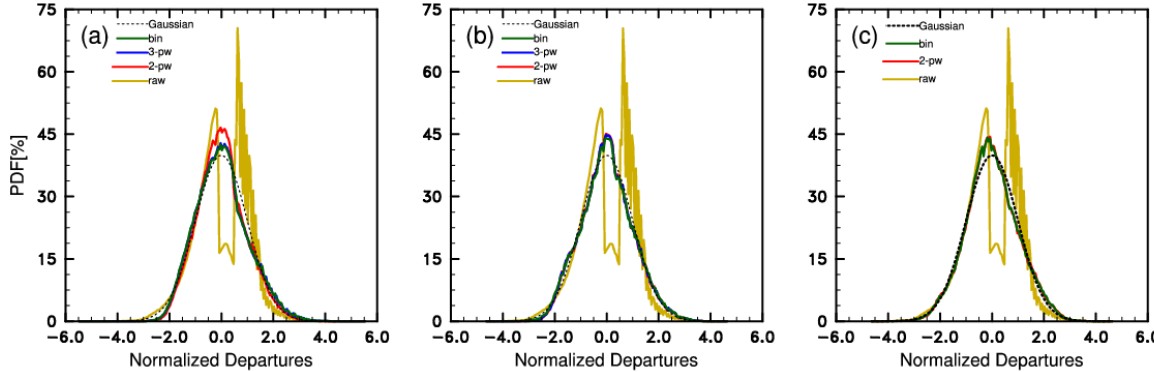

**Figure 11: Probability distribution functions (PDFs) of 'any-reflectivity' departures normalized by symmetric rainrates which are computed by (a) the CMPAS rainrates, (b) the FY-4A rainrates and (c) the 10 times logarithmic rainrates derived from reflectivity factors, respectively.**





**Table 1: Three-piecewise linear regression parameters of symmetric error models. Correlations and the root mean square errors (RMSEs) are computed by the values of fitting functions and the binned standard deviations.**

|  | function | rainrate range | correlation | RMSE |
|---|---|---|---|---|
| **derived rainrate in any-reflectivity** | y=12.13+4.12x | 0.0<x≤1.5 | 0.97 | 0.36 |
|  | y=17.63+0.77x | 1.5<x≤9.0 | 0.97 | 0.17 |
|  | y=24.55 | 9.0<x |  |  |
| **derived rainrate in both-reflectivity** | y=10.19 +3.14x | 0.0<x≤1.5 | 0.97 | 0.18 |
|  | y=14.02+0.82x | 1.5<x≤8.5 | 0.97 | 0.18 |
|  | y=20.97 | 8.5<x |  |  |
| **CMPAS in any-reflectivity** | y=12.43+2.83x | 0.0<x≤1.5 | 0.97 | 0.15 |
|  | y=16.06+0.75x | 1.5<x≤11.5 | 0.99 | 0.12 |
|  | y=24.68 | 11.5<x |  |  |
| **FY-4A in any-reflectivity** | y=14.23+0.51x | 0.0<x≤1.5 | 0.96 | 0.01 |
|  | y=14.43+0.96x | 1.5<x≤7.5 | 0.93 | 0.40 |
|  | y=21.64 | 7.5<x |  |  |

**Table 2: The same as Table 1, but for two-piecewise linear regression.**

|  | function | rainrate range | correlation | RMSE |
|---|---|---|---|---|
| **derived rainrate in any-reflectivity** | y=15.46+1.11x<br>y=25.47 | 0.0<x≤9.0<br>9.0<x | 0.93 | 1.55 |
| **derived rainrate in both-reflectivity** | y=12.43+1.08x<br>y=21.63 | 0.0<x≤8.5<br>8.5<x | 0.95 | 0.84 |
| **CMPAS in any-reflectivity** | y=14.83+0.90x<br>y=25.22 | 0.0<x≤11.5<br>11.5<x | 0.97 | 0.66 |
| **FY-4A in any-reflectivity** | y=14.03+1.03x<br>y=21.76 | 0.0<x≤7.5<br>7.5<x | 0.97 | 0.35 |
| **logarithmic rainrate in any-reflectivity** | y=15.37+0.28x<br>y=17.06 | -23.0<x≤6.0<br>6.0<x | 0.98 | 0.28 |



**Table 3: the Jensen-Shannon divergences of probability distribution functions normalized by different symmetric error models.**

|  | raw | two-piecewise | three-piecewise | binned |
|---|---|---|---|---|
| **derived rainrate in any-reflectivity** | 2.482 | 1.024 | 0.567 | 0.567 |
| **derived rainrate in both-reflectivity** | 0.263 | 0.370 | 0.137 | 0.136 |
| **CMPAS in any-reflectivity** | 2.480 | 0.442 | 0.396 | 0.400 |
| **FY-4A in any-reflectivity** | 2.479 | 0.219 | 0.214 | 0.155 |
| **Logarithmic rainrate in any-reflectivity** | 2.479 | 0.126 | - | 0.113 |