# Peer review of "Study on The Error Structure of Radar Reflectivity Using The Symmetric Rainrate Predictor"

_Atmospheric Measurement Techniques, 2023_

## Referee Comment (RC2)

Referring to the symmetric error model in satellite all-sky assimilation, this manuscript reveals the error structure of radar reflectivity as a function of symmetric rain rate, and rationally optimizes the observation error distribution based on this model. The PDF of the reflectivity departure normalized by the symmetric rain rate becomes more Gaussian compared with the PDF normalized by the original standard deviation. In addition, the effects of the predictors from different observational sources on this model are compared, which provides potential referable points for the rational estimation of observational errors in the radar data assimilation.

Major comments

1. For satellite assimilation, departures (or bias) mean O minus B (OMB), where B is calculated from the background using the observation operator in the DA model. However, the B obtained in this manuscript is calculated using the Unified Post-Processor (UPP) software package, so this B may differ from the B in the DA model. Please check the specific operator and provide an explanation.

2. Rain rates from other sources (e.g., the FY-4A QPE hourly rainfall product) have also been selected as predictors for this radar symmetric error model. However, satellite and radar are two observations of different character and perspective, please add an explanation of the rationale for this way.

3. Line 157-158: This manuscript needs to provide a description of the quality control algorithm for 'misses and false simulations', which determines the Quality of the later presentations on 'any-reflectivity' and 'both-reflectivity' analyses.

4. For the assimilation system, the symmetric error model serves to estimate the observation error at different observation points and does not change the value of the OMB, whereas the Gaussianity of Figures 10 and 11 changed. Please explain why the normalization is done by "symmetric rainrates"? Does this operation consider both the observation error and the OMB?

5. In describing the predictors for the radar symmetric error model, this manuscript has given the equal weight to the rain rate simulated by the model and the rain rate calculated from radar observations. Is it likely that the radar observations will be more accurate than the background? Please add an explanation of assigning the weights in this way.

Minor comments:

1. Line 124: The formula number is missing.

2. Line 180: Please add the strategy and time resolution for calculating $r_{model}$. In addition, the rain rate is an instantaneous variable, while the precipitation output from the WRF is an accumulated variable, and the rain rate calculated from it is an average value, please add an explanation of not using the reflectivity output directly from the WRF.

3. Line 248: Remove the 'the' before 'the Figure 7c shows that'.

4. Line 307: Replace 'more Gaussian' with 'and results in a PDF distribution that is closer to the standard Gaussian distribution'.

5. It is necessary to label the sample sizes for CMPAS rainfall, FY-4A rainfall, and the 10 times logarithmic rain rates in Figure 11.

---

## Referee Comment (RC3)

**General comments:**

**The paper uses a symmetric rain rate to define the radar reflectivity error in the assimilation algorithm based on the symmetric rain rate referring to the symmetric error model in satellite all-sky assimilation. The paper is well-structured but still, there are many ambiguous sentences in the paper which need to be rewritten/clarified.**

**major revisions:**

- The very important point that is missing in the paper is that the reflectivity error in an assimilation algorithm is a representative error based on each assimilation system/algorithm meaning that if anything changes in the assimilation algorithm either the NWP model or the number/type of observations, the reflectivity error need to be recalculated /modified. But in this paper, it seems there is not any assimilation algorithm/system. The equivalent reflectivity is based on the 6-hour model forecast which does not represent the equivalent reflectivity after assimilation. The very large standard deviation (up to 35 dbz) clearly shows this inconsistency.

- Besides, defining a better reflectivity error is supposed to improve the assimilation results. However, the paper did not show any plots related to applying the newly defined reflectivity error in an assimilation algorithm and the comparison with the constant error (which was claimed in the paper is not suitable for radar assimilation).

- Line 92: What is the purpose of defining the radar composite based on the vertical maximum reflectivity? Do you use this composite in your assimilation algorithm? Why not the radar reflectivity composites of a specific level (which will be used later in the assimilation algorithm)?

- It was mentioned that to match with the rain rate resolution of 4 or 5 km, the reflectivities with the resolution of 1 km were interpolated! Actually, this procedure is extrapolation. However, the normal trend in the radar assimilation is defining/using the super observation which defines the reflectivity over a larger grid box (compared to the original grid box) to match with other observations in the assimilation algorithm which have lower resolution.

- Fig 2: It clearly can be seen that the simulated reflectivity and the observed reflectivity are far from each other. This means the model has a very poor ability to capture the convective events. As I mentioned, this could not be the representative reflectivity of an assimilation algorithm. It can be easily seen that defining the equivalent reflectivity based on this kind of plot can cause a high value for the reflectivity standard deviation which is not realistic.

- Fig3: The two plots look identical. It couldn't/shouldn't be like this. Please check the plots.

- What is the purpose of excluding the false and missed events? and defining the 'both-reflectivity'? At the end, they all need to be included in defining the standard deviation.

- As it was written the CMPAS rain rate is the most complete one (or maybe the reference ones) what is the purpose of using another product (FY-4A)?

- In Fig 6 (or later in Fig 8), the departure or the standard deviation needs to be defined clearly. It is not clear that the departure means 'obs-model' or 'model-obs'.

- Fig 8 (Line 258): It was mentioned that the standard deviation increases after 9 mmh^-1 because of the error in the WRF model or the initial data. Actually, a big part of the error could be due to the deficiency in the forward model (the process of converting the model state to the radar reflectivity).

- Fig8: When there is (kind of) reference data set of rain rate, why the standard deviation should be plotted based on the derived rain?

- Line 279 or Line 283: What was here exactly normalized? Fig 8 and Fig 9 are the standard deviations based on the rain rate. Did you normalize any variables here?

- Fig 8: The black dashed line shows the log of sample numbers that reach less than 2 after 15 mmh^-1 meaning that the number of samples is less than 100! If this is the case, means that the number of samples in these bins is not enough to do the standard deviation. As it was shown the number of samples in some bins can reach 10^6 so there is a big inconsistency in defining the standard deviation in different bins. There should be a limit based on the number of samples to do the standard deviation. I would suggest a limit of 10^3 or 10^4.

**minor correction:**

- Line 76: "stage IV precipitation" → What is the defintion of stage IV precipitation?
- Line 99: "on May 3th 2021" → on 3rd of May 2021
- Line 218: "…, chosen to be larger than 100 samples": What does this mean? If it means that you calculate the reflectivity departure for the grid box which has more than 100 samples, why the colorbar starts from 1?
- Line 219: "excessive reflectivities" → overestimation in simulating the reflectivities
- Line 224: "It could be argued" → It shows that
- Line 235: "exists" → defines the relation between …
- Line 257: What is the "geophysical boundary"?
- Line 262: The constant reflectivity error is usually between 5 and 10 dbz
- Line 323: "… from the large mislocation errors in the main" → What is main?

---

## Author Comment (AC2)

We thank the Referee 2 for useful comments after referring previous papers about the symmetric error model. We reply all comments at length.

Major comments
1. For satellite assimilation, departures (or bias) mean O minus B (OMB), where B is calculated from the background using the observation operator in the DA model. However, the B obtained in this manuscript is calculated using the Unified Post-Processor (UPP) software package, so this B may differ from the B in the DA model. Please check the specific operator and provide an explanation.
Response:
The algorithm of diagnostic reflectivity (dBZ) included in UPP softward package can be used as an operator of reflectivity assimilation. This algorithm is based on rain, snow, and graupel mixing ratios was designed by Stoelinga (2005):

$$Z = 10 \log_{10} (Z_{er} + Z_{es} + Z_{eg}) \tag{R0}$$

Following some assumptions, the reflectivity contributed by rain droplets is given by:

$$Z_{er} = \Gamma(7)N_{r0}\lambda_r^{-7} \tag{R1}$$

$$\lambda_r = (\frac{\pi N_{r0}\rho_l}{\rho_a q_{ra}})^{1/4} \tag{R2}$$

where $N_{r0}$ is $8\times10^6$, $\rho_l$ and $\rho_a$ are the liquid water density and dry air density respectively and $q_{ra}$ is the rainwater mixing ratio in background.
Assumed snow particles are spheres, the reflectivity contributed by snow is given by:

$$Z_{es} = \alpha\Gamma(7)N_{s0}(\frac{\rho_s}{\rho_l})^2\lambda_s^{-7} \tag{R3}$$

$$\lambda_s = (\frac{\pi N_{s0}\rho_s}{\rho_a q_{sn}})^{1/4} \tag{R4}$$

where $\alpha$ is 0.224, $N_{s0}$ is $2\times10^7$, $\rho_s$ is the density of snow 100 kg m⁻³ and $q_{sn}$ is the snow water mixing ratio in background.
Similarly, the contribution of graupel particles can be obtained:

$$Z_{eg} = \alpha\Gamma(7)N_{s0}(\frac{\rho_g}{\rho_l})^2\lambda_g^{-7} \tag{R5}$$

$$\lambda_g = (\frac{\pi N_{g0}\rho_g}{\rho_a q_{gn}})^{1/4} \tag{R6}$$

where $\alpha$ is also 0.224, $N_{g0}$ is $2\times10^7$, $\rho_g$ is the density of graupel 400 kg m⁻³ and $q_{gn}$

is the graupel water mixing ratio in background.
According to above formulas (R0-R6), the reflectivity predicted by model can be computed by the rainwater, snow water and graupel water mixing ratios. It can transform model variables to reflectivity. Thus, this algorithm of diagnostic reflectivity can be used as the forward operator in reflectivity assimilation. Actually, similar forward operator of reflectivity based on double-moment Thompson

microphysics was employed by Liu et al. (2022).

Reference:
Liu, C., H. Li, M. Xue, Y. Jung, J. Park, L. Chen, R. Kong, and C. Tong, 2022: Use of a Reflectivity Operator Based on Double-Moment Thompson Microphysics for Direct Assimilation of Radar Reflectivity in GSI-Based Hybrid En3DVar. Mon. Wea. Rev., 150, 907–926, https://doi.org/10.1175/MWR-D-21-0040.1.

2. Rain rates from other sources (e.g., the FY-4A QPE hourly rainfall product) have also been selected as predictors for this radar symmetric error model. However, satellite and radar are two observations of different character and perspective, please add an explanation of the rationale for this way.
Response:
Although the geostationary satellite and ground radar are different measurements in meteorology, they can observe the same weather system and then provide similar information about the variation of convective systems. The FY-4A QPE can indicate the shape, strength and location of convective storms as rain rate retrieved by reflectivity does. Thus, the differences between satellite and radar allow us to investigate how the accuracy of predictor affects the symmetric error model. Authors explained why the third-party observations are used from line 79 to 82 and described details of third-party observations in section 3.2.

3. Line 157-158: This manuscript needs to provide a description of the quality control algorithm for 'misses and false simulations', which determines the Quality of the later presentations on 'any-reflectivity' and 'both-reflectivity' analyses.
Response:
Authors did not employ any quality control algorithm for misses and false simulations except that reflectivity less than 5 dBZ in either the observations or the simulations was excluded. The miss means an occasion where the reflectivity is observed but is not simulated. The false simulation means an occasion where the reflectivity is simulated but is not observed. By comparing the PDFs of 'any-reflectivity' and 'both-reflectivity', this study discussed what give rise to the non-Gaussian error distribution of OMBs.
To avoid possible misunderstandings, authors emphasized that the 'both-reflectivity' scenario is only used to illustrate what give rise to the non-Gaussian error distribution of radar reflectivity. In this study, the effects of more or less accurate observations and the logarithm transformation on the symmetric error model are discussed in 'any-reflectivity' scenario. Authors did not advise any reader to remove the misses and false simulations in reflectivity assimilation.

4. For the assimilation system, the symmetric error model serves to estimate the observation error at different observation points and does not change the value of the OMB, whereas the Gaussianity of Figures 10 and 11 changed. Please explain why the normalization is done by "symmetric rainrates"? Does this operation consider both the

observation error and the OMB?

Response:

As shown in Figure 8 and 9, each OMB bin, 0.5 mm h$^{-1}$ interval, is normalized separately, i.e. OMBs of reflectivity are normalized by different standard deviations. Although the PDF of all samples is not Gaussian, the PDF in each bin (a subset of all OMBs) could approximate to Gaussian. This is the heteroscedasticity of reflectivity, i.e. 'The error of equivalent reflectivity can change as a function of precipitation' as stated in Introduction. Thus, the Gaussianity can be improved because this study normalized the OMBs by using different standard deviations which are a function of rain rate. Authors would like to add some sentences in revision to explain the reason why the Gaussianityof OMBs can be improved.

5. In describing the predictors for the radar symmetric error model, this manuscript has given the equal weight to the rain rate simulated by the model and the rain rate calculated from radar observations. Is it likely that the radar observations will be more accurate than the background? Please add an explanation of assigning the weights in this way.

Response:

Changing the weights of observation and simulation of the symmetric error model (Eq. (2) in manuscript) has not been examined in this study or previous studies, and possibly it is a good idea. After carefully examining the PDF of 'any-reflectivity' (red line in Figure 4), the left and right parts of PDF show different distributions. The left part (observations less than simulations) is lower and smoother than the right part (observations larger than simulations), illustrating the PDF is not symmetric. Probably, an asymmetric predictor is the best predictor for an asymmetric PDF. It could be an interesting topic in further study.

However, the current work mainly focuses on whether the symmetric error model can improve the PDF of OMBs. According to Figure 10 and 11, the symmetric error model of radar reflectivity can improve the Gaussianity. Thus, authors would like to add a short discussion about the asymmetric predictor in revision, but not attempt to investigate the optimal weights for observation and simulation.

Minor comments:

1. Line 124: The formula number is missing.

Response: authors did not number this formula because it is not mentioned in the following manuscript.

2. Line 180: Please add the strategy and time resolution for calculating $r_{model}$. In addition, the rain rate is an instantaneous variable, while the precipitation output from the WRF is an accumulated variable, and the rain rate calculated from it is an average value, please add an explanation of not using the reflectivity output directly from the WRF.

Response: the $r_{model}$ is the average of two consecutive hourly precipitations produced by WRF. The rain rate simulated by WRF represents the simulated

3. Line 248: Remove the 'the' before 'the Figure 7c shows that'.
Response: authors can delete this word in revision.

4. Line 307: Replace 'more Gaussian' with 'and results in a PDF distribution that is closer to the standard Gaussian distribution'.
Response: authors can rewrite this sentence in revision.

---

## Author Comment (AC3)

After reading the comments from Referee 2 and Dr. Wang, authors reply the comments 2 and 4 from Referee 1 again to better communicate our thoughts. Authors would like to revise the manuscript according to all comments. Thank Referee 1 for useful comments.

2. For radar data assimilation, the volume-scan radar data are commonly used. However, the current study is based on radar reflectivity composites. One may wonder its practical value.
Response:
According to fitting functions in manuscript, the reflectivity error ($\sigma$, unit: dBZ) is inflated from a basic error, which represents the instrument error:

$$\sigma = \begin{cases} \sigma_l & RR_{avg} < RR_{avg1} \\ \sigma_l + \alpha\beta(RR_{avg} - RR_{avg1}) & RR_{avg1} \leq RR_{avg} < RR_{avg2} \\ \sigma_u & RR_{avg2} \leq RR_{avg} \end{cases}$$

where $RR_{avg}$ means the symmetric rain rate, $\sigma_l$ and $\sigma_u$ are the lower and upper boundaries of reflectivity error respectively, $\beta$ is the slope of fitting functions and $\alpha$ is a tuning parameter. The representative error of reflectivity can be described as a function of 'symmetric rain rate'. By tuning the parameter $\alpha$, the representative error can either be assigned completely by the symmetric error model ($\alpha = 1$) or ignored ($\alpha = 0$).

Authors already performed several reflectivity assimilation experiments to examine the practical value of this symmetric error model constructed by composite reflectivity. By compared with a constant error value ($\alpha = 0$), authors obtained improvements if $\alpha$ is less than 0.25, $\sigma_l$ and $\sigma_u$ are 3 and 5 dBZ respectively. Authors plan to report details of assimilation experiments in another study. It may illustrate that the heteroscedasticities of composite reflectivity and reflectivity are alike in convective systems. Because both composite reflectivity and reflectivity can provide similar information about the location, strength and shape of convective systems.

The operational centers, such as ECMWF, Met Office, NCEP and ECCC, built the symmetric error models of satellite radiance according to their own assimilation systems and prediction models. Thus, if someone wants to use the symmetric error model of radar reflectivity in practice, the symmetric error models of radar reflectivity should be built on a certain assimilation system and prediction model to obtain the appropriate inflation coefficient. The $\sigma_l$ and $\sigma_u$ also need to be discussed. To sum up, authors think this paper should focus on how to construct the symmetric error model of radar reflectivity and what impacts of the symmetric error model of radar reflectivity on Gaussianity, which may fall better in the scope of *Atmospheric Measurement Techniques*.

4. Authors should give more efforts to clarify the concept of "symmetric rainrate predictor", e.g., what does "symmetric" mean? What is advantage of it over the other methods? Throughtout the manuscript, the interpretation of results strongly rely on Geer and Bauer (2011, hereafter GB2011), which considerably reduces the relevance

of this study.

Response:

Since the major comment 4 from Referee 2 is also related to the explanation of symmetric error model, authors would like to clarify how the symmetric error model improve the Gaussianity of OMBs without changing the value of OMBs.

As shown in Figure 8 and 9, each OMB bin, 0.5 mm h$^{-1}$ interval, is normalized separately, i.e. OMBs of reflectivity are normalized by different standard deviation. Although the PDF of all samples is not Gaussian, the PDF could approximate to Gaussian in each bin. This is the heteroscedasticity of reflectivity, i.e. 'The error of equivalent reflectivity can change as a function of precipitation' as stated in Introduction. Thus, the Gaussianity can be improved because this study normalized the OMBs by using different standard deviations which are a function of rain rate.

---

## Author Comment (AC4)

It is an interesting study introducing a method designed for satellite data to estimate radar data errors. However, there are some statements that need clarification.

Thank Dr. Wang for constructive comments. Authors leave responses in blue words.

1. According to the abstract, the purpose of this work is unclear. Did the author aim to estimate the reflectivity error or rainrate error? What is the innovation of the present work? Will the present work provide referential information for data assimilation (DA)? All of these points should be stated explicitly.
Response:
This study used the symmetric error model, which is widely used in all-sky satellite radiance assimilation, to unveil the heteroscedasticity of composite reflectivity, which is rarely discussed before.
The fitting functions in manuscript could be used to inflate the observation error of radar reflectivity, which may represent the representative error in reflectivity assimilation:

$$\sigma = \begin{cases} \sigma_l & RR_{avg} < RR_{avg1} \\ \sigma_l + \alpha\beta(RR_{avg} - RR_{avg1}) & RR_{avg1} \leq RR_{avg} < RR_{avg2} \\ \sigma_u & RR_{avg2} \leq RR_{avg} \end{cases} \qquad (R1)$$

where $RR_{avg}$ means the symmetric rain rate, $\sigma_l$ and $\sigma_u$ are the lower and upper boundaries of reflectivity error respectively, $\beta$ is the slope of fitting functions and $\alpha$ is a tuning parameter. By tuning the parameter $\alpha$, the representative error can either be assigned completely by the symmetric error model ($\alpha = 1$) or ignored ($\alpha = 0$).
However, the symmetric error model cannot be a general function for all data assimilation system and prediction model. It has to be built according to a certain data assimilation system and prediction model. Thus, authors think this paper should focus on how to construct the symmetric error model of radar reflectivity and what impacts of the symmetric error model of radar reflectivity on Gaussianity, which may fall better in the scope of *Atmospheric Measurement Techniques*.

2. The statement "The error of equivalent reflectivity can change as a function of precipitation" raises the question if the precipitation mentioned involves ice phase hydrometers. If it does, why is rainrate used in the abstract instead of reflectivity? Additionally, why should the error be symmetric? No related context is provided before this.
Response:
The essential point of earlier symmetric error models in all-sky microwave radiance assimilation is that the hydrometeor predictor is derived from the radiances themselves, either the observations or the equivalent radiance simulations. In this study, authors use derived rainrate to construct the symmetric error model for composite reflectivity. Both composite reflectivity and rainrate are associated with the location, shape and strength of convective systems, which is the sources of representative error in reflectivity assimilation. Thus, authors unveiled the structure of

representative error by using the symmetric error model.

Essentially, describing the heteroscedasticity of the ice phase hydrometers needs a three-dimensional symmetric predictor. However, how to design a three-dimensional predictor to describe the variation of reflectivity errors in three-dimensional space is still a challenge. Authors stated this issue at line 375.

The name symmetric error model is from Geer and Bauer (2011). The symmetric refers to the average of observations and simulations. The predictor only computed by background has a bias. By contrast, the symmetric predictor gives a PDF that is closer to Gaussian, as shown by Figure 6 in Geer and Bauer (2011).

Reference:
Geer, A. J., and Bauer, P.: Observation errors in all-sky data assimilation. *Q J R Meteorol Soc*, 137, 2024-2037, https://doi.org/10.1002/qj.830, 2011.

3. How can we exclude the impact of ice phase particles when estimating rainrate using radar reflectivity in terms of the Z-I relationship?
Response:

Authors employed the Z-I relationship to compute the predictor of symmetric error model of composite reflectivity. The results of CMPAS data sets, which is a more accurate observation than derived rainrate, are similar to the results of derived rainrate. Authors may argue that the Z-I relationship with classical coefficients is accurate enough to compute the symmetric predictor.

4. Again, in the introduction, I understand what the authors planned to do, but I'm unclear about the purpose. The motivation should have been clearer.
Response:

Authors appreciate so much time two anonymous referees and Dr. Wang spent and will revise seriously the manuscript according to all comments from them.

5. In this study, according to the symmetric error model constructed by the rainrate predictor, the standard deviations of reflectivity could range from about 12 to 35 dBZ. Should we believe the authors' claims that the error is indeed so large?"
Response:

How to use the fitting functions possibly is described in the response of comment 1. The observation error of reflectivity assimilation could be limited to $10^0$ order by the lower and upper boundaries in Eq. R1. Authors will clarify the possible usage in revision.

---

## Author Comment (AC5)

**General comments:**

The paper uses a symmetric rain rate to define the radar reflectivity error in the assimilation algorithm based on the symmetric rain rate referring to the symmetric error model in satellite all-sky assimilation. The paper is well-structured but still, there are many ambiguous sentences in the paper which need to be rewritten/clarified.

Authors appreciate the constructive comments from referee #3. Many operation centers, such as ECMWF, NCEP, ECCC and CMA, have reported the positive impacts of using the symmetric error model in all-sky satellite radiance assimilation. Although those symmetric error models have been built on the basis of different NWP models and assimilation systems, even different symmetric predictors, authors noticed that the procedure for building the symmetric error model is the same. Moreover, the reflectivity assimilation suffers similar issues in all-sky satellite radiance assimilation. Thus, authors demonstrated the symmetric error model can attack the non-Gaussian problem in radar reflectivity assimilation.

After carefully considering the comments left by all referees, including previous reviewers from *Earth and Space Science*, authors think this study better entitles "*Improving the Gaussianity by Using the Symmetric Rain Rate Toward Reflectivity Assimilation*" and hope this study could engage the readers who can build the symmetric error model based on their own NWP models and assimilation systems.

Authors reply all comments from Referee #3 and explained our ideas in the following blue words.

**major revisions:**

1• The very important point that is missing in the paper is that the reflectivity error in an assimilation algorithm is a representative error based on each assimilation system/algorithm meaning that if anything changes in the assimilation algorithm either the NWP model or the number/type of observations, the reflectivity error need to be recalculated /modified. But in this paper, it seems there is not any assimilation algorithm/system. The equivalent reflectivity is based on the 6-hour model forecast which does not represent the equivalent reflectivity after assimilation. The very large standard deviation (up to 35 dbz) clearly shows this inconsistency.

Response:

Authors agree the comment that the representative error in reflectivity assimilation is highly related to a certain NWP model and assimilation algorithm. The symmetric error model reported by this study may be inappropriate if the NWP system is changed. It is the reason that this study mainly focuses on how to build the symmetric error model based on observation and simulation data sets, which is rarely discussed before and may fall better in the scope of *Atmospheric Measurement Techniques*. Authors also expect that this study engages radar assimilation experts who want to build their symmetric error models by following the procedure reported in this study.

For the large inconsistency between observations and simulations, authors would like to brief the usage of the symmetric error model. Similar to the all-sky satellite

radiance assimilation, the reflectivity error (σ, unit: dBZ) is inflated from a lower boundary to an upper boundary:

$$\sigma = \begin{cases} \sigma_l & RR_{avg} < RR_{avg1} \\ \sigma_l + \alpha\beta(RR_{avg} - RR_{avg1}) & RR_{avg1} \leq RR_{avg} < RR_{avg2} \\ \sigma_u & RR_{avg2} \leq RR_{avg} \end{cases} \quad (R1)$$

where $RR_{avg}$ means the symmetric rain rate, $\sigma_l$ and $\sigma_u$ are the lower and upper boundaries of reflectivity error respectively, $\beta$ is the slope of fitting functions and $\alpha$ is a tuning parameter. By tuning the parameter $\alpha$, the representative error can either be assigned completely by the symmetric error model ($\alpha = 1$) or ignored ($\alpha = 0$). Tuning the parameter $\alpha$, as designed by Geer and Bauer (2011), can improve applicability of the symmetric error model. Besides, tens of dBZ of reflectivity departures between observations and 6 hour forecast is common for convective NWP model.

Reference:
Geer, A. J., and Bauer, P.: Observation errors in all-sky data assimilation. *Q J R Meteorol Soc*, 137, 2024-2037, https://doi.org/10.1002/qj.830, 2011.

2• Besides, defining a better reflectivity error is supposed to improve the assimilation results. However, the paper did not show any plots related to applying the newly defined reflectivity error in an assimilation algorithm and the comparison with the constant error (which was claimed in the paper is not suitable for radar assimilation).
Response:
  According to the PDF distributions (Fig. 10 and 11) and JSD (Table 3) in this manuscript, the symmetric error model can improve the raw non-Gaussian distribution of reflectivity error. At least in theory, a more Gaussian distribution is more consistent with most current data assimilation algorithms. Moreover, applying the symmetric error model on reflectivity assimilation could be very complicated, because some empirical parameters in Eq. (R1) should be discussed by data assimilation experiments. Authors will add the Eq. (R1) in our revision in order to clarify how to apply the symmetric error model on reflectivity. Considering the length and innovation, authors will not add any data assimilation experiment and still focus on how to build the symmetric error model and its impacts on PDF.

3• Line 92: What is the purpose of defining the radar composite based on the vertical maximum reflectivity? Do you use this composite in your assimilation algorithm? Why not the radar reflectivity composites of a specific level (which will be used later in the assimilation algorithm)?
Response:
  Compared with the radar reflectivity composites of a specific level, the vertical maximum reflectivity not only represents the strength of convections, but provides more samples. Authors do not assimilate the vertical maximum reflectivity.
  Although the definitions of composite reflectivity and rain rate have distinct

differences, they are good indicators of convective storms. The strength and distribution of composite reflectivity and rain rate are associated with the variation of convective systems. The following Figure R1 shows the absolute correlations between the composite reflectivity and various rain rate data sets in the six months. For the derived rain rate data sets, most cases show the absolute correlations are about 0.75, despite some cases present low correlations. Thus, we argue that the composite reflectivity and derived rain rate are comparable for a precipitating weather system and the derived rain rates can be used to describe the heteroscedasticity of composite reflectivity in statistics, similar to the cloud liquid water or liquid water path for satellite radiances.

[Figure]

Figure R1. the absolute correlations between the composite reflectivity and various rain rate data sets in the six months. The black, blue and red lines represent the rain rate derived from reflectivity at 3 km altitude, the CMPAS data sets and the FY-4A data sets. The dash line shows the 95% confidence.

The absolute correlations of CMPAS and FY-4A decrease obviously because the independent errors from the third-party data sets, including the sampling and representative errors, increase rapidly. However, all averaged absolute correlations, which are 0.62, 0.35 and 0.24 for derived rain rate, CMPAS and FY-4A data sets respectively, can pass the 95% confidence. We argue that the differences among the three rain rate data sets allow us to investigate how the accuracy of predictor affects the symmetric error model.

[Figure]

Figure R2. Probability distribution functions (PDFs) of OmBs (observations minus backgrounds)

at 1 km altitude normalized by the standard deviation of the whole samples (green line) and by the symmetric error model (red line). the black line represents the normal Gaussian PDF.

Authors also employed the radar reflectivity composites at 1 km altitude to investigate the PDF as shown in Fig. R2. Compared with Fig. R4, the PDFs of 1 km reflectivity are similar to those of the vertical maximum reflectivity composite. It illustrates that the symmetric error models of the vertical maximum reflectivity and the radar reflectivity composites of a specific level are similar. Authors also found the symmetric error model built by the radar reflectivity composites at 1 km altitude is similar to that built by the vertical maximum reflectivity.

Thus, the piecewise function (Eq. R1) fitted by the vertical maximum reflectivity and the derived rain rate can be used to inflate the observation error at some specific levels, which are lower than the melting level.

4 • It was mentioned that to match with the rain rate resolution of 4 or 5 km, the reflectivities with the resolution of 1 km were interpolated! Actually, this procedure is extrapolation. However, the normal trend in the radar assimilation is defining/using the super observation which defines the reflectivity over a larger grid box (compared to the original grid box) to match with other observations in the assimilation algorithm which have lower resolution.

Response:

It should be interpolation because the domain with 1 km resolution is larger than the domain with 5 km resolution. The values in coarse grids (red dash lines) were computed by the adjacent four points in fine grids (black solid lines), as shown in Figure R2. This data thinning strategy is similar to the super observation.

[Figure]

Figure R3. the schematic of thinning 1 km reflectivity.

This comment reminds authors that the zero reflectivity is usually not assimilated. For rainy echo, reflectivities larger than 5 dBZ are assimilated. Thus, the reflectivites smaller than 5 dBZ should be excluded when computing the PDF and building the symmetric error model. The Fig. R4 shows the PDF of OmBs of the vertical maximum composite reflectivity. Compared to the PDF of 'any-reflectivity' in Fig. 4, the PDF (green line) becomes an unimodal distribution. The negative peak disappeared because observations smaller than 5 dBZ are removed. Authors think this PDF distribution is closer to the reality in current reflectivity. The high peak is reduced if OmBs normalized by the symmetric model (red line). All reflectivity data sets are updated and authors will re-investigate the symmetric error model in revision.

[Figure]

Figure R4. Probability distribution functions (PDFs) of OmBs (observations minus backgrounds) of the vertical maximum composite reflectivity normalized by the standard deviation of the whole samples (green line) and by the symmetric error model (red line). the black line represents the normal Gaussian PDF.

5 • Fig 2: It clearly can be seen that the simulated reflectivity and the observed reflectivity are far from each other. This means the model has a very poor ability to capture the convective events. As I mentioned, this could not be the representative reflectivity of an assimilation algorithm. It can be easily seen that defining the equivalent reflectivity based on this kind of plot can cause a high value for the reflectivity standard deviation which is not realistic.

Response:

Nowadays, many researches show near perfect simulations, but distinct disagreements between observations and simulations, such as the poor shape, strength and location, are common cases in daily operation. Because the NWP model and its initial data have limited ability to describe the distributions and variations of hydrometeors. The OmBs (observations minus backgrounds) often increase to tens of dBZ in 6 hours simulation, let alone the error of reflectivity operator.

Admittedly, using poor simulations exaggerates the representative error. The tuning parameter $\alpha$ in Eq. (R1) may remedy it by reducing the slope of fitting function $\beta$.

6 • Fig3: The two plots look identical. It couldn't/shouldn't be like this. Please check the plots.

Response:

The two plots are different on the abscissa and ordinate, which illustrate the impact of misses and false simulations on PDF in Fig. 4. However, according to the fourth comment, authors will re-do the construction of symmetric error model using the observations larger than 5 dBZ in the revision. Because it is more consistent with the reality of reflectivity assimilation.

7• What is the purpose of excluding the false and missed events? and defining the 'both reflectivity'? At the end, they all need to be included in defining the standard deviation.

Response:

Excluding the false and missed events is used to illustrate what give rise to the non-Gaussian error distribution. To avoid misunderstandings, authors will remove all plots about the 'both reflectivity' from Section 3. The construction of symmetric error model will be only based on the 'any reflectivity' in revision.

8• As it was written the CMPAS rain rate is the most complete one (or maybe the reference ones) what is the purpose of using another product (FY-4A)?

Response:

The essential point of earlier symmetric error models in all-sky microwave radiance assimilation is that the hydrometeor predictor is derived from the radiances themselves, either the observations or the equivalent radiance simulations. Thus, the derived rain rate is the reference in this study.

Authors attempted to use third-party rain rate data sets to replace the derived rain rate data sets, which is a major different from previous works. As shown in Fig. R1, the CMPAS rain rate is more accurate, but its correlation is lower than the derived rain rate. Because the third-party rain rate has its own representation and mislocation errors. However, authors attempted to argue that the third-party rain rate could be still useful in construction of symmetric error model, if the independent error of the third-party is small. the third-party rain rate cannot be used if the independent error is large, which is the case of FY-4A.

Authors can remove the plots and discussions related to FY-4A data sets since it is far away the aim of this study.

9• In Fig 6 (or later in Fig 8), the departure or the standard deviation needs to be defined clearly. It is not clear that the departure means 'obs-model' or 'model-obs'.

Response:

All departures are OmB (observations minus backgrounds). Authors will clearly define it in revision.

10• Fig 8 (Line 258): It was mentioned that the standard deviation increases after 9 mm h$^{-1}$ because of the error in the WRF model or the initial data. Actually, a big part of the error could be due to the deficiency in the forward model (the process of converting the model state to the radar reflectivity).

Response:

Authors agree this comment. The error of observation operator could become large when ice-phased hydrometeors exist. The observation operator is single moment, which is based on rain, snow, and graupel mixing ratios was designed by Stoelinga (2005):

$$Z = 10 \log_{10} \left( Z_{er} + Z_{es} + Z_{eg} \right) \tag{RR0}$$

Following some assumptions, the reflectivity contributed by rain droplets is given by:

$$Z_{er} = \Gamma(7)N_{r0}\lambda_r{}^{-7} \tag{RR1}$$

$$\lambda_r = (\frac{\pi N_{r0}\rho_l}{\rho_a q_{ra}})^{1/4} \tag{RR2}$$

where $N_{r0}$ is $8\times10^6$, $\rho_l$ and $\rho_a$ are the liquid water density and dry air density respectively and $q_{ra}$ is the rainwater mixing ratio in background.

Assumed snow particles are spheres, the reflectivity contributed by snow is given by:

$$Z_{es} = \alpha\Gamma(7)N_{s0}(\frac{\rho_s}{\rho_l})^2\lambda_s{}^{-7} \tag{RR3}$$

$$\lambda_s = (\frac{\pi N_{s0}\rho_s}{\rho_a q_{sn}})^{1/4} \tag{RR4}$$

where $\alpha$ is 0.224, $N_{s0}$ is $2\times10^7$, $\rho_s$ is the density of snow 100 kg m$^{-3}$ and $q_{sn}$ is the snow water mixing ratio in background.

Similarly, the contribution of graupel particles can be obtained:

$$Z_{eg} = \alpha\Gamma(7)N_{s0}(\frac{\rho_g}{\rho_l})^2\lambda_g{}^{-7} \tag{RR5}$$

$$\lambda_g = (\frac{\pi N_{g0}\rho_g}{\rho_a q_{gn}})^{1/4} \tag{RR6}$$

where $\alpha$ is also 0.224, $N_{g0}$ is $2\times10^7$, $\rho_g$ is the density of graupel 400 kg m$^{-3}$ and $q_{gn}$

is the graupel water mixing ratio in background. This algorithm can be used as the forward operator in reflectivity assimilation. Similar forward operator of reflectivity based on double-moment Thompson microphysics was employed by Liu et al. (2022). In future, the impacts of observation operator on the symmetric error model should be discussed.

Authors will add some sentences in revision to illustrate the deficiency in the forward model

Reference:
Liu, C., H. Li, M. Xue, Y. Jung, J. Park, L. Chen, R. Kong, and C. Tong, 2022: Use of a Reflectivity Operator Based on Double-Moment Thompson Microphysics for Direct Assimilation of Radar Reflectivity in GSI-Based Hybrid En3DVar. Mon. Wea. Rev., 150, 907–926, https://doi.org/10.1175/MWR-D-21-0040.1.

11• Fig8: When there is (kind of) reference data set of rain rate, why the standard deviation should be plotted based on the derived rain?
Response:
Same to the fifth comment, the derived rain rate is the reference data set in the symmetric error model. The CMPAS and FY-4A data sets were used to discuss the impacts of the third-party observations on the symmetric error model.

12• Line 279 or Line 283: What was here exactly normalized? Fig 8 and Fig 9 are the standard deviations based on the rain rate. Did you normalize any variables here?
Response:

In Fig. 8 and Fig. 9, each OmB bin, 0.5 mm $h^{-1}$ interval, is normalized separately, i.e. OmBs of reflectivity are normalized by different standard deviations, instead of normalization by the standard deviation of the whole samples.

This is the artful normalization to attack the non-Gaussian PDF. Although the PDF of whole samples is not Gaussian, the whole samples can be separated by many sub-groups with Gaussian PDF. Thus, the observation errors in reflectivity can be estimated separately.

13• Fig 8: The black dashed line shows the log of sample numbers that reach less than 2 after 15 mm $h^{-1}$ meaning that the number of samples is less than 100! If this is the case, means that the number of samples in these bins is not enough to do the standard deviation. As it was shown the number of samples in some bins can reach $10^6$ so there is a big inconsistency in defining the standard deviation in different bins. There should be a limit based on the number of samples to do the standard deviation. I would suggest a limit of $10^3$ or $10^4$.
Response:

Authors could not agree this comment more. The linear regression is lower than 9 mm $h^{-1}$, where the sample number is larger than $10^3$. Thus, the piecewise function is a constant when the sample number is smaller than $10^3$. Authors will emphasize this point in revision.

**minor correction:**
• Line 76: "stage IV precipitation" → What is the defintion of stage IV precipitation?
Response: the stage IV precipitation is a precipitation data set produced by NCEP. We mentioned it here to support the high correlation between reflectivity and precipitation.

• Line 99: "on May 3th 2021" → on 3rd of May 2021
Response: authors will correct this mistake in revision.

• Line 218: "…, chosen to be larger than 100 samples": What does this mean? If it means that you calculate the reflectivity departure for the grid box which has more than 100 samples, why the colorbar starts from 1?
Response: this sentence attempted to describe that the results is credible when the samples larger than 100. Authors will revise this sentence in revision.

• Line 219: "excessive reflectivities" → overestimation in simulating the reflectivities
Response: authors will rewrite this sentence in revision.

- Line 224: "It could be argued" → It shows that

Response: authors will rewrite this sentence in revision.

- Line 235: "exists" → defines the relation between ⋯

Response: authors will rewrite this sentence in revision.

- Line 257: What is the "geophysical boundary"?

Response: it means the boundary in the map, such as the boundary between rainy and non-rainy areas.

- Line 262: The constant reflectivity error is usually between 5 and 10 dbz

Response: the constant reflectivity error is empirical. Authors have reviewed several constants in different researches. Authors will rewrite 5 or10 dBZ in revision.

- Line 323: "⋯ from the large mislocation errors in the main" → What is main?

Response: this sentence attempted to explain the large mislocation errors are the main reason of non-Gaussian distribution. Authors will rewrite this sentence in revision.